# The Proteomic Analysis of Cancer-Related Alterations in the Human Unfoldome

**DOI:** 10.3390/ijms25031552

**Published:** 2024-01-26

**Authors:** Victor Paromov, Vladimir N. Uversky, Ayorinde Cooley, Lincoln E. Liburd, Shyamali Mukherjee, Insung Na, Guy W. Dayhoff, Siddharth Pratap

**Affiliations:** 1Meharry Proteomics Core, RCMI Research Capacity Core, School of Medicine, Meharry Medical College, Nashville, TN 37208, USA; vparomov@mmc.edu; 2Department of Molecular Medicine, USF Health Byrd Alzheimer’s Research Institute, Morsani College of Medicine, University of South Florida, Tampa, FL 33613, USA; vuversky@usf.edu (V.N.U.); nais8303@gmail.com (I.N.); 3Meharry Bioinformatics Core, Department of Microbiology, Immunology and Physiology, School of Medicine, Meharry Medical College, Nashville, TN 37208, USA; acooley@mmc.edu; 4Department of Biochemistry, Cancer Biology, Neuroscience & Pharmacology, School of Medicine, Meharry Medical College, Nashville, TN 37208, USAsmukherjee@mmc.edu (S.M.); 5Department of Chemistry, College of Art and Sciences, University of South Florida, Tampa, FL 33613, USA; gdayhoff@usf.edu

**Keywords:** unfoldome, intrinsically disordered proteins (IDPs), intrinsically disordered protein regions (IDPRs)

## Abstract

Many proteins lack stable 3D structures. These intrinsically disordered proteins (IDPs) or hybrid proteins containing ordered domains with intrinsically disordered protein regions (IDPRs) often carry out regulatory functions related to molecular recognition and signal transduction. IDPs/IDPRs constitute a substantial portion of the human proteome and are termed “the unfoldome”. Herein, we probe the human breast cancer unfoldome and investigate relations between IDPs and key disease genes and pathways. We utilized bottom-up proteomics, MudPIT (Multidimensional Protein Identification Technology), to profile differentially expressed IDPs in human normal (MCF-10A) and breast cancer (BT-549) cell lines. Overall, we identified 2271 protein groups in the unfoldome of normal and cancer proteomes, with 148 IDPs found to be significantly differentially expressed in cancer cells. Further analysis produced annotations of 140 IDPs, which were then classified to GO (Gene Ontology) categories and pathways. In total, 65% (91 of 140) IDPs were related to various diseases, and 20% (28 of 140) mapped to cancer terms. A substantial portion of the differentially expressed IDPs contained disordered regions, confirmed by in silico characterization. Overall, our analyses suggest high levels of interactivity in the human cancer unfoldome and a prevalence of moderately and highly disordered proteins in the network.

## 1. Introduction

Many functional proteins lack stable 3D structures as a whole or in their substantial parts due to specific and discoverable features at the primary amino acid sequence level. These intrinsically disordered proteins (IDPs) and intrinsically disordered protein regions (IDPRs) often carry out regulatory functions and are related to molecular recognition, regulation, and signal transduction, with disorder-based functionality being complementary to the common catalysis and transport activities of proteins with well-defined, stable three-dimensional structures [1,2,3,4,5,6,7,8,9,10,11,12,13,14,15,16,17,18,19,20,21,22,23,24,25,26]. Therefore, protein functionality may originate from both order and disorder, and the classic structure-function paradigm, which emphasizes that ordered 3D structures represent the indispensable prerequisite to effective protein functioning, should be redefined to include IDPs/IDPRs [1,3,4,6,8,9,12,13,26,27].

Structurally, IDPs range from completely unstructured polypeptides and extended partially structured forms to compact disordered ensembles containing substantial levels of secondary structures [3,4,6]. Furthermore, many proteins are order-disorder hybrids, containing a mix of ordered domains and IDPRs. These hybrid proteins contain regions with different degrees and flavors of disorder and in general, are characterized by a remarkable degree of structural heterogeneity. This very complex and heterogeneous spatiotemporal structural organization of a protein molecule includes foldons (independent foldable units of a protein), inducible foldons (disordered regions that can fold at least in part due to interactions with binding partners), morphing inducible foldons (disordered regions that can fold into different structures upon interaction with different partners), non-foldons (non-foldable protein regions), semi-foldons (regions that are always in a semi-folded form), and unfoldons (ordered regions that have to undergo an order-to-disorder transition to become functional) [28]. In its turn, this complex structural organization defines the unique functionality of IDPs/IDPRs, where differently (dis)ordered structural elements might have well-defined and specific functions, thereby defining the possibility of a protein molecule being multifunctional [28]. This also defines the relationships of protein structure and function in the form of a structure-function continuum concept, where instead of the classical “one gene–one protein–one structure–one function” view, the actual protein structure-function relationship is described by the more convoluted “one-gene–many-proteins–many-functions” model [28]. 

Since IDPs/IDPRs constitute a substantial portion of any known proteome [29,30,31,32], have amazing structural variability, and possess a very wide variety of functions, the concepts of unfoldome and unfoldomics have been introduced [13]. Here, unfoldomics represents a sub-field of protein science that studies the unfoldome, which includes a set of intrinsically disordered proteins (also known as natively unfolded proteins, therefore unfoldome), their functions, structures, interactions, evolution, etc. [13]. Furthermore, intrinsic disorder is tightly connected to the pathogenesis of various human diseases, since the majority of human cancer-associated proteins [7,33,34,35,36,37,38,39,40]—as well as many proteins associated with neurodegeneration [41,42,43,44,45,46,47], diabetes [48,49,50,51], major psychiatric disorders [52], and cardiovascular diseases [48,53,54,55,56,57]—are either intrinsically disordered or contain long IDPRs, giving rise to the D^2^ (disorder in disorders) concept [58]. Curiously, based on the computational analysis of the diseasome network organized by human genetic diseases and related genes [59], it was concluded that this diseasome represents a human-genetic-disease-associated unfoldome, where intrinsic disorder and disorder-based interaction sites are commonly found in proteins associated with human genetic diseases and where proteins from different classes of genetic disease possess different levels of intrinsic disorder [60]. It was also emphasized that disease-associated mutations are commonly found within IDPRs [54].

Based on the systematic comparison of the peculiarities of the amino acid sequences of ordered proteins/domains and IDPs/IDPRs, it was found that IDPs/IDPRs differ from ordered globular proteins and domains at many levels. Among the most notable differences are dissimilarities in amino acid compositions, sequence complexity, hydrophobicity, charge, and flexibility [3,12,61]. At a more detailed level, IDPs/IDPRs were shown to be systematically depleted in order-promoting residues (Cys, Trp, Ile, Tyr, Phe, Leu, His, and Val), being noticeably enriched in disorder-promoting amino acids (Pro, Ala, Glu, Ser, Gln, Lys, Gly, Asp, and Arg) [3,12,61]. 

Furthermore, the lack of compact structures in highly disordered IDPs/IDPRs (i.e., IDPs with so-called extended disorders, such as coil-like and pre-molten globule-like) was attributed to the characteristic combination of low mean hydrophobicity and relatively high net charge [2]. These peculiarities in the amino acid sequences of extended IDPs define their “turned out” response to heat, where increases in temperature cause the formation of more ordered secondary structures [62,63,64]. These structure-promoting effects of elevated temperatures were attributed to the increased strength of hydrophobic interactions at higher temperatures, leading to stronger hydrophobic attractions, which is the major driving force for folding [64]. Analogously, extended IDPs show a “turned out” response to changes in pH [64,65,66,67], where partial folding was shown to be induced in extended IDPs by a decrease (or increase) in pH of the media. This pH-induced partial folding of extended IDPs/IDPRs was attributed to the minimization of their large net charge at a neutral pH, leading to a decrease in intramolecular electrostatic repulsion and thereby permitting the hydrophobicity-driven collapse of the partially folded conformation [28,64]. 

Importantly, it was emphasized that the aforementioned structural features of extended IDPs and their specific conformational behavior can be utilized in the large-scale identification of these important members of the protein kingdom, where the resistance of such proteins to acid- or heat-treatment can be utilized for their targeted identification [68,69,70]. In fact, one such method is based on the finding that many proteins that fail to precipitate during perchloric acid or trichloroacetic acid treatment are IDPs [68], whereas another method utilizes the fact that IDPs possess high resistance toward heat-induced aggregation [68,69,70].

Therefore, since IDPs/IDPRs are often associated with various diseases, including cancer, we suggest that a focused unfoldome characterization would represent a valuable addition to cancer discovery research. To this end, here, we probe a human breast cancer unfoldome generated by the treatment of human normal (MCF-10A) and breast cancer (BT-549) cells with 1.5% trichloroacetic acid, followed by IDP precipitation and an LC-MS/MS analysis, and investigated the relations of these IDPs to key disease genes and pathways in silico using bioinformatics approaches.

## 2. Results and Discussion

### 2.1. Proteomics Analysis

Utilizing a bottom-up proteomics approach and Multidimensional Protein Identification Technology (MudPIT) methods [71,72,73,74,75,76,77,78,79,80,81], we compared the unfoldomics portions of normal (MCF-10A cell lysates) and cancerous (BT-549 cell lysates) proteomes. In general, IDPs show altered physical properties in solution under physiological conditions—e.g., higher resistance to temperature, light, or pH-induced precipitation [64,68]. To remove folded proteins, we treated the cell lysates with 1.5% trichloroacetic acid and sedimented protein pellets by centrifugation. Next, we sedimented all the proteins using 20% trichloroacetic acid, and thus, we obtained a fraction of a proteome highly enriched in IDPs. Our MudPIT procedure identified 2271 protein groups in these unfoldomic fractions (Figure 1 and Figure 2). The unfoldomic enrichment revealed substantial differences between normal and cancer unfoldomes: 148 IDPs were significantly up or downregulated in cancer cells (≥|2 − fold|; *p* ≤ 0.05). The further analysis of these differentially present proteins (DPPs) allowed annotation of 140 proteins using the Database for Annotation, Visualization, and Integrated Discovery knowledgebase (DAVID) (v6.8, accessed on 12/2023) [82,83,84]. Notably, 65% (91 of 140) IDPs were related to various diseases, and 20% (28 of 140) of IDPs were related to cancer (GAD_DISEASE database). These data suggest the substantial alteration of the human unfoldome in cancer.

### 2.2. In Silico Protein Disorder Analysis

Figure 3 provides a representative characterization of the 140 DPPs identified in this study. Although the proteins assembled in the human cancer unfoldome range in length from 68 (Copper transport protein ATOX1, UniProt ID: O00244) to 5871 residues (nesprin-2 isoform X12, RefSeq ID: XP_011534885.1), Figure 3A that shows the length distribution of these proteins demonstrates that most identified proteins are rather short (less than 400 residues). Since amino acid compositions of ordered and intrinsically disordered proteins are known to be noticeably different, with IDPs possessing lower levels of order-promoting (mostly hydrophobic) residues and higher levels of polar and charged residues [3,12,61], we utilized the Composition Profiler, a tool for the detection of enrichment or depletion patterns of individual amino acids within query proteins. The results (summarized in Figure 3B) clearly show that proteins in the human cancer unfoldome are depleted in most order-promoting residues and enriched in most disorder-promoting residues, highly suggesting that these proteins contain significant levels of intrinsic disorder. 

The validity of this conclusion is supported by Figure 4, which represents the outputs of different intrinsic disorder predictors for a set of query proteins. Since the average disorder score (ADS) of a given protein is not directly related to its percent of predicted disordered residues (PPDR) value, Figure 4A shows the PPDR vs. ADS plots generated for human proteins that are differently expressed in cancer based on the output of Rapid Intrinsic Disorder Analysis Online (RIDAO) [89]. RIDAO yields results for IUPred (short), IUPred (long), PONDR^®^ VL3, PONDR^®^ VLXT, PONDR^®^ VSL2, and PONDR^®^ FIT and computes a mean disorder score for each residue based on these predictors. The plots illustrate a reasonable agreement between the outputs of different intrinsic disorder predictors for the analyzed proteins. These data can also be used for the classification of proteins by their overall disorder characteristics. In fact, two arbitrary cutoffs for the levels of intrinsic disorder (PPDR values) are typically used to classify proteins as highly ordered (PPDR < 10%), moderately disordered (10% ≤ PPDR < 30%), and highly disordered (PPDR ≥ 30%) [90]. Similarly, proteins can be classified based on their ADS values, with proteins being considered as highly ordered, moderately disordered, and highly disordered if their ADS values are <0.25, between 0.25 and 0.5, or ≥0.5, respectively. Therefore, Figure 4A demonstrates that the majority of analyzed DPPs are either moderately or highly disordered. Furthermore, this analysis allowed us to select the most disordered proteins in the set, which were defined here as proteins possessing ADS ≥ 0.5 or PPDR ≥ 30% (by any of the predictors utilized in this study). We found that 103 (69.6%) and 77 (52.0%) of the DPPs satisfy these criteria, respectively.

Figure 4B compares two more conservative ways of evaluation of intrinsic disorder—PPDRs evaluated by the meta-predictor PONDR^®^ FIT and the RIDAO mean disorder score—and shows the presence of reasonable correlation between these approaches. 

Next, we compared the outputs of PONDR^®^ VLXT, PONDR^®^ VSL2, and PONDR^®^ FIT in the form of a 3D scatter plot, where the PPDR values generated by these three disorder predictors are plotted against each other for all the DPPs (Figure 4C). Based on this analysis and the aforementioned PPDR thresholds, 21, 51, and 76 members of the human cancer unfoldome are ordered, moderately disordered, and highly disordered proteins, respectively (see Figure 4C). In other words, according to these criteria, 85.8% of these proteins are predicted to possess noticeable levels of intrinsic disorder. Similarly, we created a 3D scatter plot correlating the corresponding ADS values (Figure 4D). Based on this analysis and the aforementioned ADS thresholds, 29, 72, and 47 human proteins differently expressed in cancer are expected to be ordered, moderately disordered, and highly disordered, respectively (see Figure 4D), suggesting that by these criteria, 80.4% proteins contain high levels of disorder. 

Important information on the global classification of disorder status of query proteins can be obtained from the analysis of binary disorder predictors (i.e., computational tools that classify proteins as wholly ordered or wholly disordered). Because of the principal differences in the criteria used by two such predictors, the charge-hydropathy (CH) plot [2,31] and the cumulative distribution function (CDF) plot [31], their combination in the form of a CH-CDF plot gives an opportunity for the unique assessment of intrinsic disorder in several categories, allowing for the predictive classification of proteins into structurally different classes [31]. The results of this analysis are shown in Figure 5A, where based on their positions within the CH-CDF phase space: proteins are classified as ordered (proteins predicted as ordered and compact by both the CDF and CH, which are located within the lower-right quadrant (Q1)), native molten globules, or hybrid proteins containing sizable levels of order and disorder (proteins predicted to be disordered by the CDF but compact by the CH-plot, which are located within the lower-left quadrant (Q2)); proteins with extended disorder, such as native coils and native pre-molten globules (proteins predicted to be disordered by both methods, which are found within the upper-left quadrant (Q3)); and proteins predicted to be disordered by the CH-plot but ordered by the CDF (located within the upper-right quadrant (Q4)) [31]. 

This analysis revealed that Q1 includes 71 DPPs, of which 21 proteins are predicted to be highly ordered, 40 proteins classified as moderately disordered, and 10 proteins classified as highly disordered by PONDR^®^ FIT. Q2 includes 30 DPPs (3 moderately and 27 highly disordered proteins according to the PONDR^®^ FIT analysis), whereas in Q3, one can find a total of 38 proteins, of which 2 and 36 are predicted as moderately and highly disordered (respectively) by PONDR^®^ FIT. Finally, Q4 contains 8 DPPs, of which 4 and 4 were classified as moderately and highly disordered (respectively) by PONDR^®^ FIT. Figure 5B further illustrates these observations in a 3D scatter plot, where in addition to the CH-CDF plane, the PONDR^®^ FIT-generated PPDR values are used as a third dimension. Overall, these analyses revealed that more than half of the proteins altered in the human cancer unfoldome are globally disordered. 

### 2.3. Biological Network and Pathway Analyses

These DPPs (140 protein groups) were classified using major Gene Ontology (GO) categories, and a pathway analysis was performed using an interacting genes database (DAVID) [82,83,84]. For insight into the potential function of the comorbid signature proteins, we performed a protein-protein interaction (PPI) analysis of the identified DPPs in each group comparison using GeneMANIA [91] and with previously established bioinformatics workflows from our lab [92].

First, we analyzed the inter-set interactivity of these DPPs using the Search Tool for the Retrieval of Interacting Genes (STRING (http://string-db.org/, accessed on 20 December 2023)) [93,94,95]. Figure 6 represents the results of this analysis for 146 DPPs (no STRING information was available for XP_005264440.1 and XP_011528471.1) and shows that most of these proteins are involved in the formation of a rather dense PPI network. In fact, only two of the proteins (TMA16 and TMEM263 with UniProt IDs: Q96EY4 and Q8WUH6, respectively) are not included in this network. As a result, this network contains 145 nodes (human proteins from the cancer unfoldome) connected by 1603 edges (PPIs). In this network, the average node degree is 22.1, and its average local clustering coefficient (which defines how close its neighbors are to being a complete clique; the local clustering coefficient is equal to 1 if every neighbor connected to a given node *N_i_* is also connected to every other node within the neighborhood, and it is equal to 0 if no node that is connected to a given node *N_i_* connects to any other node that is connected to *N_i_*) is 0.422. Since the expected number of interactions among proteins in a comparable size set of proteins randomly selected from the human proteome is equal to 769, this internal PPI network has significantly more interactions than expected, being characterized by a PPI enrichment *p*-value < 10^−16^. Therefore, these data indicate that the majority of human proteins differently expressed in cancer can interact with each other (although the confidence of this inter-family PPI network is low).

Next, we compared the interactivity of highly ordered (PPDR < 10%), moderately disordered (10% ≤ PPDR < 30%), and highly disordered (PPDR ≥ 30%) DPPs. Figure 7 shows the interaction networks for the 125 most-related proteins out of 140 annotated DPPs (excluding 8 unknown proteins), which are grouped together in clusters.

Figure 8 and Figure 9 show the same comparison of interactivity for the DPPs related to cancer and diseases other than cancer (respectively) according to the DAVID functional annotation. The cancer PPI network consists of 21 (+12 singletons) proteins, where 13 (+9 singletons) proteins are moderately or highly disordered, and 7 (+3 singletons) proteins are ordered.

The moderately disordered heat shock protein 90 alpha family class A (HSP90AA1) interacted with the most proteins in the network. HSP00AA1 has roles in tumor formation and activates oncogenic proteins to enhance tumor growth and invasiveness [96,97]. The highly disordered protein with the most PPIs was stress induced phosphoprotein 1 (STIP1). STIP1 overexpression is associated with high metastatic potential in diverse types of cancer [98,99,100]. Heat shock protein family A, member 1A (HSPA1A), was also highly connected. In colon cancer patients, elevated levels of HSPA1A are associated with a poor prognosis [101]. Among the other disordered proteins, Keratin 5 (KRT5) is a biomarker in basal-like breast cancers associated with a poor prognosis [102].

Several ordered proteins were highly connected. Heat shock 70 kDA protein 5 (HSPA5) has been shown to promote tumor growth by regulating cell proliferation in various cancers [103,104]. Phosphoglycerate kinase 1 (PGK1) has roles in angiogenesis and DNA repair. PGK1 is overexpressed in many cancers, and its overexpression typically correlates with poor prognoses in specific cancer types. However, PGK1 can act as a tumor suppressor by suppressing angiogenesis and tumor growth under certain conditions [105]. Different post-translational modifications of PGK1 are related to the positive or negative regulation of tumor metabolism and growth [106].

Superoxide dismutase 2 is an antioxidant that regulates oxidative stress in the mitochondria through the dismutation of superoxide radicals. The upregulation of SOD2 contributes to cancer cell proliferation, angiogenesis, and invasion [107,108] or tumor suppression [107]. The stability of the SOD2 protein observed in this study is consistent with findings indicating the SOD2 gene is rarely mutated in cancer. It acts as a context-dependent regulator of tumorigenesis or tumor suppression [107].

The network of annotations for other diseases consisted of 48 proteins. In total, 34 proteins were moderately or highly disordered, and 14 proteins were ordered. The highly disordered protein Vinculin (VCL) had the most PPIs. VCL is responsible for the progression of different cancers through tumor cell invasion and proliferation [109,110]. In prostate cancer patients, the down-regulation of the VCL promoter leads to the repression of tumor invasion, movement, and migration [110]. The human ribosomal large subunit protein L7a (RPL7A) contributes to the risk of breast cancer with adequate alcohol consumption. Specifically, the protein functions as an ethanol-response factor in breast cancer cells and promotes tumor growth and metastasis [111]. NME1-NME2 is capable of suppressing metastasis in several types of cancers [112,113]. Actinin alpha 4 (ACTN4) is extensively associated with cancer development. Both in vivo and in vitro studies have implicated this protein in tumor invasion and metastasis [114,115]. Depending on the cellular context, ACTN4 may also function as a tumor suppressor [116].

Finally, we analyzed the pathway enrichment for the 140 annotated DPP using the Kyoto Encyclopedia of Genes and Genomes (KEGG) pathways and associated proteins database (https://www.genome.jp/kegg/, accessed on 20 December 2023) [117,118,119,120,121]. We observed significant enrichment of protein processing in the endoplasmic reticulum pathway (04141, *adj_p* = 0.0014). Cancer cells frequently undergo ER stress, triggering the unfolded protein response (UPR) to promote cell survival [122]. HSP90AA1, a component of this pathway, may serve as a target of interest due to its extensive interactions with ordered and disordered proteins. We observed the enrichment of numerous pathways corresponding to cell adhesion, junctions, and extracellular matrix function. Functional defects in these structures are typically associated with increased metastatic potential by the promotion of epithelial-mesenchymal transition [123,124]. Alterations in spliceosome pathway (03040, *adj_p* = 0.0315) components have been associated with aggressive forms of cancer, promoting cell invasion, angiogenesis, and drug resistance [125].

The antigen processing and presentation pathway (04612, *adj_p* = 0.0087) plays a significant role in both tumor evasion and the development of cancer immunotherapies. IDPs may have a role in the functions of antigen-presenting machinery in tumor cells, possibly interfering with the antitumor response. Proteasome activator subunit 2 (PSME2, PA28) is constitutively expressed in antigen-presenting cells and is involved in MHC class I antigen presentation. The down-regulation of this protein has been observed to impair the antigen presentation of endogenous tumor antigens [126,127]. Disorder in PSME2 protein may also impair its activity and inhibit the antigen-presenting response.

## 3. Materials and Methods

### 3.1. Cell Culture

Human normal (MCF-10A) and breast cancer (BT-549) cell lines were cultured in Dulbecco’s Modified Eagle’s medium (DMEM) with 2 mmol/L L-glutamine and 10% fetal bovine serum (VWR). Upon confluence, cells were trypsinized, sedimented at 1000× *g* for 6 min, and stored at −80 °C.

### 3.2. Sample Preparation

Prior to the LC-MS analysis, cell pellets (~10^7^ cells per sample) were lysed on ice with 0.1% Nonidet P-40 in PBS in the presence of the Protease Inhibitor Cocktail (Sigma-Aldrich) for 30 min. Cell debris were removed by centrifugation at 13,000× *g* for 30 min. Then, fully folded proteins were precipitated at 13,000× *g* for 30 min after incubation with 1.5% trichloroacetic acid (TCA). The pellets were removed and discarded. Then, IDPs were precipitated at 13,000× *g* for 60 min after incubation with 20% TCA. The IDP containing pellets (50 μg total protein per sample) were denatured in 8M urea and 50 mM Tris-HCl, pH 8.0, reduced with 10 mM TCEP for 60 min, alkylated with 2 mM iodoacetamide for 60 min in the dark, and then diluted to 2M urea with 50 mM Tris-HCl, pH 8.0, at RT. Two micrograms of Trypsin Gold (Promega) were added for overnight digestion (18 h, 37 °C), and then the tryptic peptides were immediately desalted using Pierce C18 spin columns (Thermo Fischer Scientific, Norcross, GA USA) at RT. Peptides were eluted with 70% acetonitrile and 0.1% formic acid (FA) and dried completely on a SpeedVac Concentrator.

### 3.3. Nano-LC-MS/MS Analysis

Peptides were resuspended in 5 μL of 0.5% FA and loaded onto a three-phase MudPIT column (150 μm × 2 cm C18 resin, 150 μm × 4 cm strong cation exchange SCX resin, filter union, and 150 μm × 20 cm C18 resin) as described previously [128]. A 10-step MudPIT (0 mM, 25 mM, 50mM, 100 mM, 150 mM, 200 mM, 300 mM, 500 mM, 750 mM, and 1000 mM ammonium acetate, each salt pulse followed by a 120-min acetonitrile gradient 5–50% B [Buffer A: 0.1% FA; Buffer B: 0.1% FA in acetonitrile]) was executed for the LC-MS analysis using an Eksigent™ AS-1 autosampler and Eksigent™ nano-LC Ultra 2D pump online with an Orbitrap LTQ XL Linear Ion Trap Mass Spectrometer (Thermo Finnigan, Norcross, GA USA) with a nanospray source. MS data acquisition was undertaken with a data-dependent six-event method (a survey FTMS scan [res. 30,000] followed by five data-dependent IT scans for the five consequent most abundant ions). The general mass spectrometric settings were as follows: spray voltage, 2.4 kV; no sheath and no auxiliary gas flow; ion transfer tube temperature, 200 C; CID fragmentation (for MS/MS), 35% normalized collision energy; activation q = 0.25; activation time, 30 ms. The minimal threshold for the dependent scans was set to 1000 counts, and a dynamic exclusion list was used with the following settings: repeat count of 1, repeat duration of 30 s, exclusion list size of 500, and exclusion duration of 90 s.

### 3.4. Proteins Identification and Quantification

Database searches were undertaken using PEAKS v9 software (Bioinformatics Solutions Inc., Waterloo, ON, Canada) against the forward and reverse human trypsin sequences (UniProtKB/Swiss-Prot) [85,129,130,131]. The parameters for the database search were as follows: full tryptic digestion; up to three missed cleavage sites; 20 ppm peptide mass tolerance; 0.5 Da fragment mass tolerance; cysteine carbamidomethylation (+57 Da) and methionine oxidation (+16 Da) as variable modifications. The relative label-free quantification (LFQ) of the identified proteins was performed with the Q module of the PEAKS software based on the extracted ion currents of the identified unique peptides’ parent ions or a spectral counting approach, and statistically significant changes have been confirmed with Fisher’s exact test (*p* ≤ 0.005; Benjamini-Hochberg FDR < 0.05).

### 3.5. Protein Annotation, Biological Network and Pathway Analyses

Annotation in DAVID (https://david.ncifcrf.gov/, accessed on 20 December 2023) [82,83,84] was conducted to identify proteins associated with cancer and other non-cancer diseases. Biological networks were constructed with the Cytoscape software platform (https://cytoscape.org/, accessed on 20 December 2023) [132] and visualized with GEPHI (https://gephi.org/, accessed on 20 December 2023) [133]. The pathway enrichment analysis was performed with WEBGESTALT web analysis software (http://bioinfo.vanderbilt.edu/wg2/, accessed on 20 December 2023) [134,135] by mapping proteins corresponding to KEGG pathways and conducting a hypergeometric test for significant enrichment. The significance for pathway level enrichment was defined as having an enrichment score False Discovery Rate (FDR) adjusted *p*-value ≤ 0.05 (*adj_p* ≤ 0.05). 

### 3.6. Compositional Profiling of the Members of Human Cancer-Related Unfoldome

A comparative analysis of the amino acid composition of the members of the human cancer-related unfoldome with respect to DisProt 3.4 (http://www.disprot.org, accessed on 20 December 2023) [88] and PDB Select 25 [87] reference protein sets was conducted using the Composition Profiler online service (http://www.cprofiler.org, accessed on 20 December 2023) [86]. The set DisProt 3.4 comprises consensus sequences of experimentally determined disordered regions, while PDB Select 25 contains PDB structures with less than 25% sequence identity and is biased towards the composition of proteins amenable to crystallization studies. Amino acids are arranged in the order of increase in their disorder propensity, according to the scale by Radivojac et al. [12].

### 3.7. Per-Residue Intrinsic Disorder Analysis

The per-residue intrinsic disorder predisposition analysis of proteins included in the unfoldome of human cancer and proteins included in the human cancer-related un-foldome was conducted using the RIDAO platform [89], which integrates a suite of six established disorder predictors: PONDR^®^ VLXT [136], PONDR^®^ VL3 [137], PONDR^®^ VLS2B [138], PONDR^®^ FIT [139], IUPred (short), and IUPred (long) [140,141]. RIDAO enables the rapid generation of disorder profile plots for individual polypeptides as well as arrays of polypeptides. We note that the use of multiple computational tools for prediction of intrinsic disorder in proteins is an accepted practice in the field. This is because different computational tools use different attributes (such as amino acid composition, hydropathy, sequence complexity, etc.) and models to calculate a disorder predisposition score for every amino acid residue in a query protein. Therefore, it is not uncommon for different tools to generate rather different outputs. Furthermore, there is not an accepted consensus regarding which disorder predictor is the best for evaluating disorder predisposition of a query protein. In fact, since different computational tools are sensitive to different disorder-related aspects of amino acid sequences, all of them contain some useful information. The per-residue disorder propensities that these tools outputs are real numbers between 1 (ideal prediction of disorder) and 0 (ideal prediction of order). To identify the disordered residues and regions in query proteins, a threshold of ≥0.5 is used. For each query protein in this study, the predicted percentage of intrinsic disorder (PPID) was calculated based on the outputs of each predictor in the RIDAO suite, such that PPID in a query protein represents the percent of residues with disorder scores satisfying the aforementioned threshold.

### 3.8. Binary and CH-CDF Intrinsic Disorder Analysis

Next, RIDAO was used to perform a CH-CDF analysis [142]. A CH-CDF analysis takes into consideration the outputs of two binary predictors—i.e., the charge-hydropathy (CH) plot [2,31] and the cumulative distribution function (CDF) plot [31]. RIDAO yields CH values based on the Kyte-Doolittle hydropathy scale [143] and CDF values based on PONDR^®^ VLXT [136]. Conducting a CH-CDF analysis allows for the classification of proteins based on their position within the CH-CDF phase space as: (i) ordered (proteins predicted to be ordered by both binary predictors), (ii) putative native “molten globules” or hybrid proteins (proteins determined to be ordered/compact by the CH but disordered by the CDF), (iii) putative native coils and native pre-molten globules (proteins predicted to be disordered by both methods), and (iv) proteins predicted to be disordered by the CH-plot but ordered by the CDF.

### 3.9. Evaluation of the Interactability within the Human Cancer-Related Unfoldome

Information on the interactability within the human cancer-related unfoldome was retrieved using the Search Tool for the Retrieval of Interacting Genes; STRING (http://string-db.org/, accessed on 20 December 2023) [93,94,95]. STRING generates a network of protein-protein interactions based on predicted and experimentally validated information on the interaction partners of a protein of interest [93,94,95]. In the corresponding network, the nodes correspond to proteins, whereas the edges show predicted or known functional associations. Seven types of evidence are used to build the corresponding network, where they are indicated by differently colored lines: a green line represents neighborhood evidence; a red line—the presence of fusion evidence; a purple line—experimental evidence; a blue line—co-occurrence evidence; a light blue line—database evidence; a yellow line—text mining evidence; and a black line—co-expression evidence [93].

In this study, STRING was utilized to generate the inter-unfoldome network of protein-protein interactions (PPI), where the protein interaction network was obtained from STRING using a custom confidence level (minimum required interaction score) of 0.15. We also analyzed the individual PPI networks of the 10 most disordered proteins from the analyzed set using a custom value of 500 maximum first-shell interactions at medium, high, and highest confidence levels (minimum required interaction score of 0.4, 0.7, and 0.9 respectively). The resulting PPI networks were further analyzed using STRING-embedded routines in order to retrieve network-related statistics, such as the following: the number of nodes (proteins); the number of edges (interactions); average node degree (average number of interactions per protein); average local clustering coefficient (which defines how close the neighbors are to being a complete clique—if a local clustering coefficient is equal to 1, then every neighbor connected to a given node *N_i_* is also connected to every other node within the neighborhood, and if it is equal to 0, then no node that is connected to a given node *N_i_* connects to any other node that is connected to *N_i_*); expected number of edges (which is the number of interactions among the proteins in a random set of proteins of similar size); and a PPI enrichment *p*-value (which is a reflection of the fact that query proteins in the analyzed PPI network have more interactions among themselves than what would be expected for a random set of proteins of similar size drawn from the genome; it has been pointed out that such an enrichment indicates that the proteins are at least partially biologically connected as a group).

## 4. Conclusions

The results reported in this study confirmed and further characterized the role of IDPs in the molecular mechanisms of breast cancer. The involvement of IDPs in several fundamental cellular processes underscores the need to capture their distinct contributions to cancer. Understanding the structural dynamics of these IDPs and the mechanics of their binding interactions with other proteins can provide new insights into the role of protein dysregulation in cancer development. As approaches for characterizing IDP structures and their interactors develop, the unique characteristics of specific cancer phenotypes can be elucidated.

## Figures and Tables

**Figure 1 ijms-25-01552-f001:**
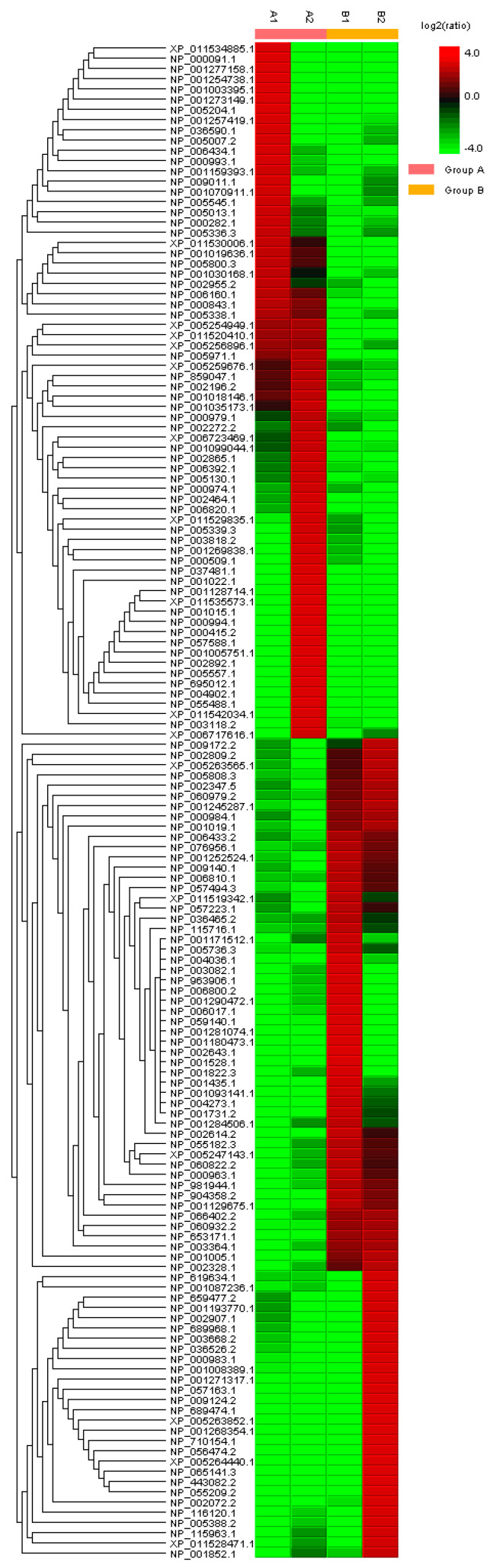
Heat map with hierarchical clustering of the 148 DPPs in the unfoldomics portions of the normal (MCF-10A cell lysates) versus cancerous (BT-549 cell lysates) human proteomes using PEAKS Studio proteomics software (PEAKS Version 9, accessed on 12/2023) [85]. Clustering: rows are centered; unit variance scaling is applied to rows. Both rows and columns are clustered using correlation distance and average linkage. For the heat map, the legend on the right indicates the color key for intermediate ratios: red color = up-regulated proteins; green color = down-regulated proteins; color gradation according to the log2 ratio (as shown).

**Figure 2 ijms-25-01552-f002:**
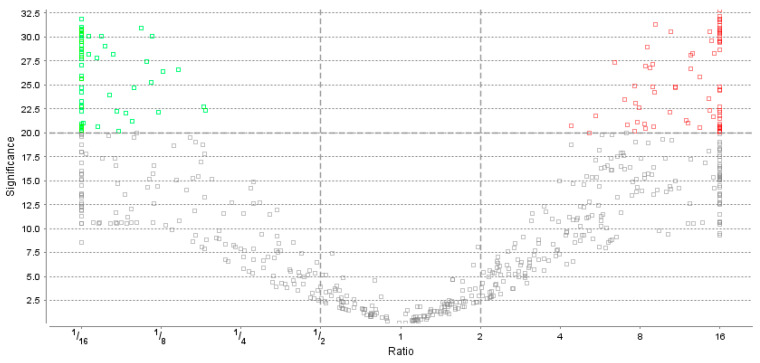
Volcano plot showing distribution of the up-regulated (red) and down-regulated (green) proteins in the unfoldome portions of the normal (MCF-10A cell lysates) and cancerous (BT-549 cell lysates) human proteomes. Filtering parameters: PEAKS significance score ≥ 20 or −10_logP > 2 (equivalent to *p* ≤ 0.01) and fold change ≥ |2|. (PEAKS Version 9, accessed on 12/2023).

**Figure 3 ijms-25-01552-f003:**
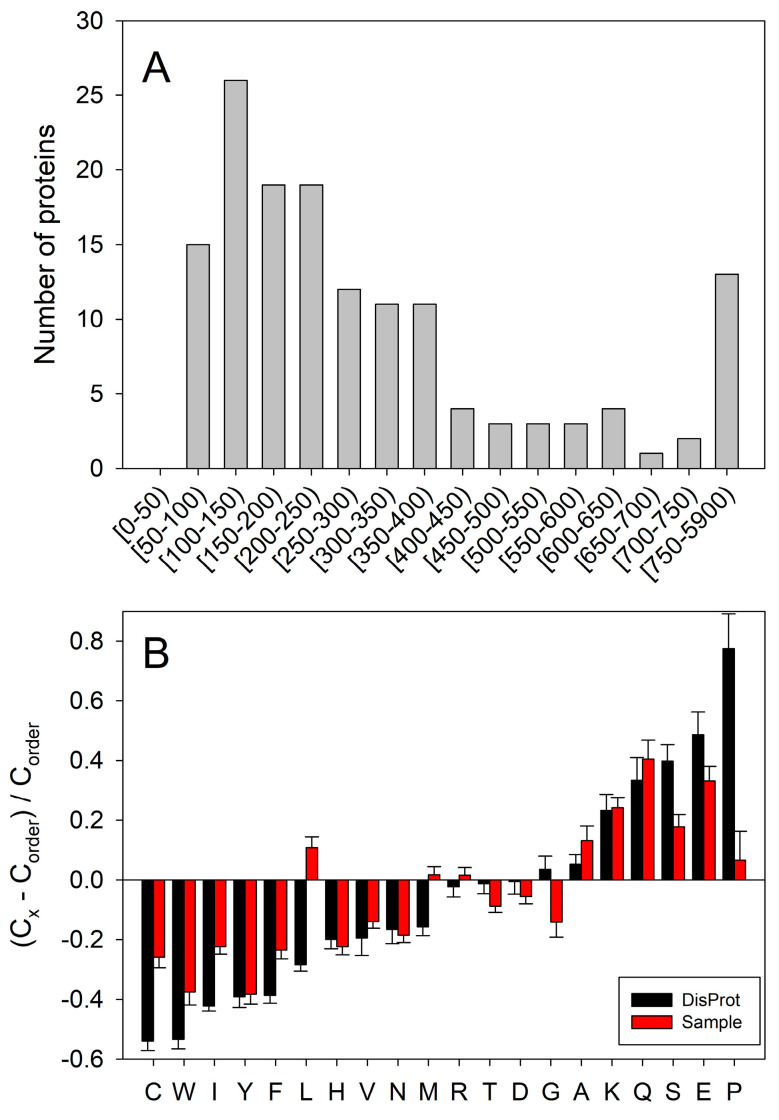
Characteristics of 140 human proteins differently expressed in the cancer unfoldome. (**A**) Length distribution of proteins. (**B**) Compositional profiling of proteins conducted using the web-based computation tool, Composition Profiler (http://profiler.cs.ucr.edu, accessed on 20 December 2023) [86]. This tool generates fractional compositional differences calculated as (C − Corder)/Corder, where C is the content of a given amino acid in a query protein (set), and Corder is the corresponding value for the set of ordered proteins from PDB Select 25 (a subset of all PDB proteins sharing ≤ 25% sequence identity) [87]. The magnitudes of the bars indicate fractional differences between amino acid compositions, where positive/negative values indicate enrichment/depletion, respectively. Error bars indicate standard deviations of fractional differences based on Composition Profiler bootstrapping. Black bars correspond to a similar analysis for proteins via DisProt [88].

**Figure 4 ijms-25-01552-f004:**
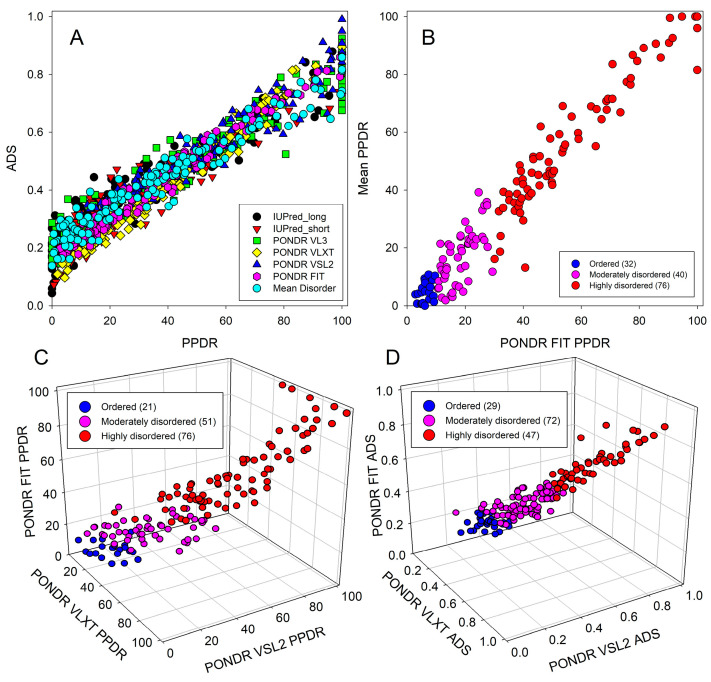
Two- and three-dimensional scatterplots comparing the average disorder scores and average PPDR values generated by three different predictors. (**A**). Average disorder score (ADS) vs. predicted percent disordered residues (PPDR) for the entire RIDAO suite of predictors and the RIDAO mean disorder score. (**B**). PPDR based on RIDAO’s mean disorder score vs. PPDR based on PONDR^®^ FIT. (**C**). PPDR evaluated by PONDR^®^ VLXT, PONDR^®^ VSL2, and PONDR^®^ FIT along the x, y, and z axes, respectively. (**D**). ADS generated by PONDR^®^ VLXT, PONDR^®^ VSL2, and PONDR^®^ FIT along the x, y, and z axes, respectively. Proteins shown in blue, pink, or red are predicted as high ordered, moderately disordered, and highly disordered, respectively, based on their respective PONDR^®^ FIT PPDR values.

**Figure 5 ijms-25-01552-f005:**
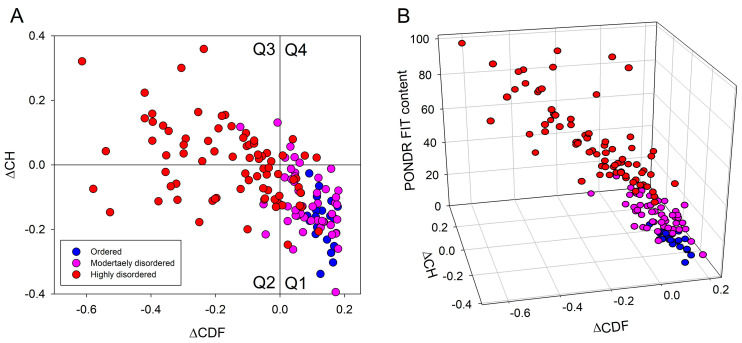
Evaluation of the overall disorder status of 148 human proteins differently expressed in cancer. (**A**). CH-CDF plot analysis of 166 human TPR proteins. Quadrants are numbered in a clockwise direction starting with Q1 in the lower right and ending with Q4 in the upper right. The following characterize proteins within each quadrant: Q1 (lower-right quadrant)—proteins predicted to be ordered by both predictors; Q2 (lower-left quadrant)—proteins predicted to be ordered by CH but disordered by CDF; Q3 (upper-left quadrant)—proteins predicted to be disordered by both predictors; and Q4 (upper-right quadrant)—proteins predicted to be disordered by CH and ordered by CDF. (**B**). Comparison of the CH-CDF plane with the output of the per-residue disorder predictor PONDR^®^ FIT.

**Figure 6 ijms-25-01552-f006:**
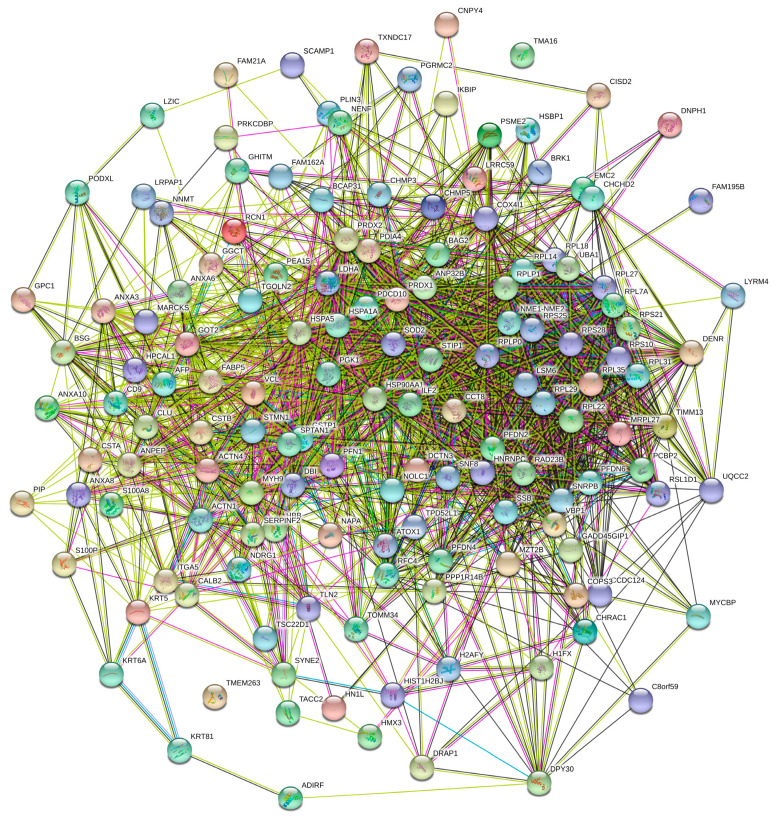
STRING-based analysis of the inter-set interactivity of 140 human proteins differently expressed in cancer using the low confidence level of 0.15. This confidence level was selected to ensure maximal inclusion of these proteins in the resulting PPI. STRING generates a network of predicted associations based on predicted and experimentally validated information on the interaction partners of a protein of interest [93]. In the corresponding network, the nodes correspond to proteins, whereas the edges show predicted/known functional associations. Seven types of evidence are used to build the corresponding network and are indicated by the differently colored lines: a green line represents neighborhood evidence; a red line—the presence of fusion evidence; a purple line—experimental evidence; a blue line—co-occurrence evidence; a light blue line—database evidence; a yellow line—text mining evidence; and a black line—co-expression evidence [93].

**Figure 7 ijms-25-01552-f007:**
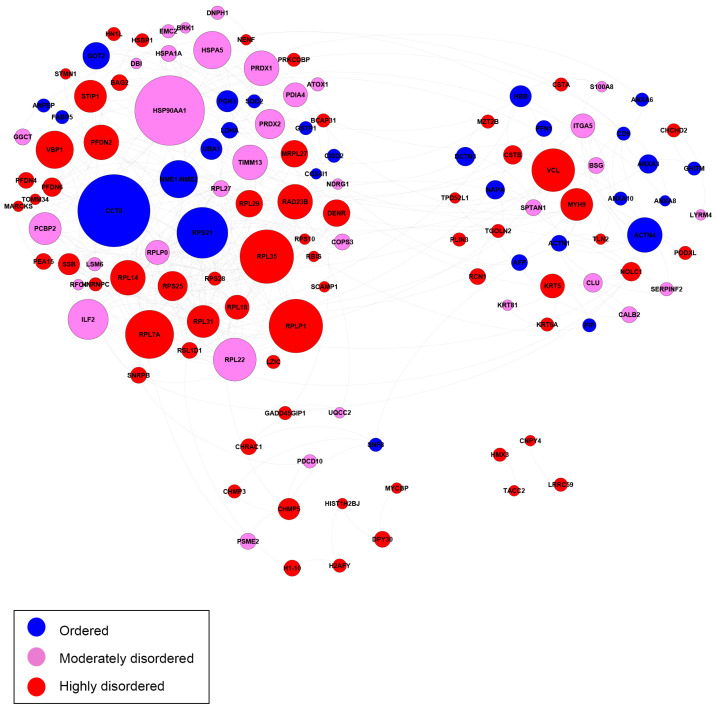
Biological interaction networks of the differentially present proteins (DPPs). In total, 125 most-related proteins (a subgroup of 148 DPPs excluding 8 unknown proteins) are shown and grouped together in clusters. Node size is proportional to the number of connections to other nodes. Nodes are colored according to the levels of intrinsic disorder (PPDR values): highly ordered, PPDR < 10%; moderately disordered, 10% ≤ PPDR < 30%; and highly disordered, PPDR ≥ 30% (see the explanation in Section 3.

**Figure 8 ijms-25-01552-f008:**
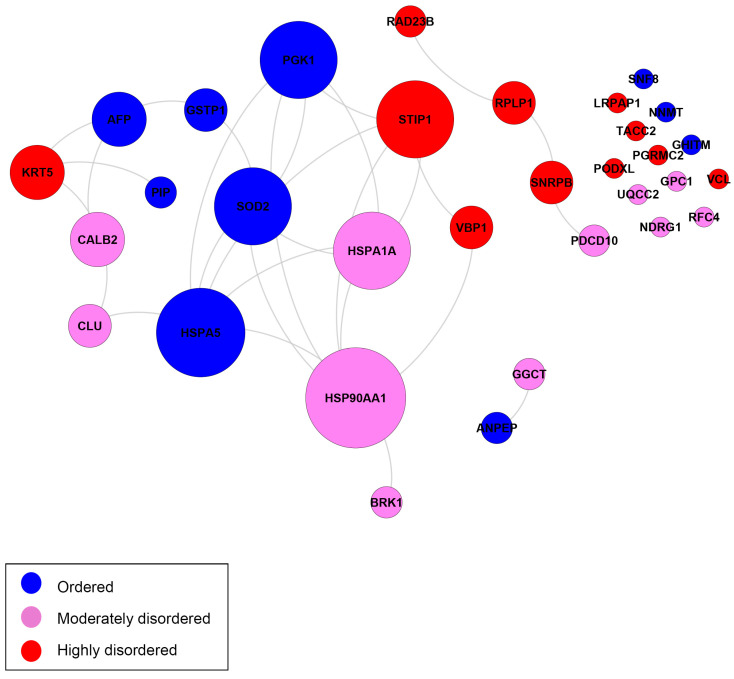
Biological interaction networks of the DPPs related to cancer (a subgroup of 148 DPPs) according to the DAVID functional annotation. In total, 20 most-related proteins out of 28 cancer related proteins are shown and grouped together in clusters. Node size is proportional to the number of connections to other nodes. Nodes are colored according to the levels of intrinsic disorder (PPDR values): highly ordered, PPDR < 10%; moderately disordered, 10% ≤ PPDR < 30%; and highly disordered, PPDR ≥ 30% (see the explanation in Section 3).

**Figure 9 ijms-25-01552-f009:**
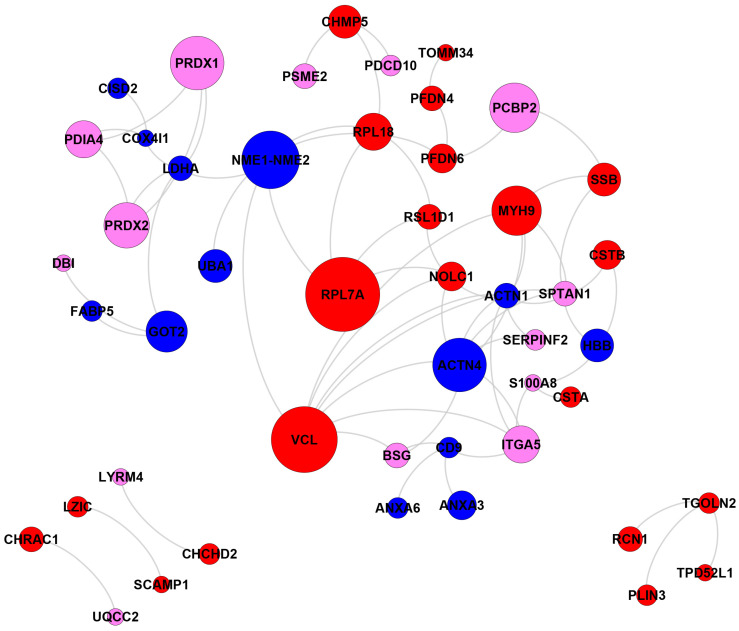
Biological interaction networks of the DPPs related to diseases other than cancer (a subgroup of 148 DPPs) according to DAVID functional annotation (GAD_DISEASE). In total, 49 most-related proteins out of 63 disease related proteins (other than cancer) are shown and grouped together in clusters. Node size is proportional to degree (the number of connections to other nodes). Nodes are colored according to the levels of intrinsic disorder (PPDR values): highly ordered, PPDR < 10%; moderately disordered, 10% ≤ PPDR < 30%; and highly disordered, PPDR ≥ 30% (see the explanation in Section 3).

## Data Availability

The mass spectrometry proteomics data have been deposited to the ProteomeXchange Consortium via the PRIDE partner repository with the dataset identifier PXD047284 and 10.6019/PXD047284 (https://www.proteomexchange.org, accessed on 1 January 2024).

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
