# Peer review of "The Proteomic Analysis of Cancer-Related Alterations in the Human Unfoldome"

_ijms, 2024, doi:10.3390/ijms25031552_

Round 1
Reviewer 1 Report
Comments and Suggestions for Authors
The authors compared unfoldomes between normal and maligant cells, and found 148 proteins are significantly up or downregulated in cancer cells. Since intrinsic disordered protein (IDP) is relatively stable at low concentration trichloricacetic acid treatment, they were thought to be IDP or intrinsically disordered protein regions (IDPRs). The in-silico characterization supported the idea.
Since 65% (91 of 140 annotated proteins) were related to 31 various diseases, and 20% (28 of 140) mapped to cancer terms. The authors concluded that there is high levels of interactivity in the human cancer unfoldome and a revalence of moderately and highly disordered proteins in the network.
It is interesting study. However, the authors should revise the manuscript according to the following queries.
1. I am interested in the difference of up-regulated and down-regulated IDPs/IDPRs in the malignant cells. The authors should give the detailed discussion on it.
2. It is unclear that the proteomic difference in unflodome is significant compared with the difference in the total proteome. The authors should give a statistical comparison.
3. Fig.3B shows that proteins in the human cancer unfoldome are depleted in most order-promoting residues and enriched in most disorder-promoting residues. I tells only that the cancer unfoldome has almost same tendency of amino acid composition. It is meaningless.
Reviewer 2 Report
Comments and Suggestions for Authors
The authors of this article are studying the development of breast cancer in humans, and also explore the association of IDPs with key genes and disease pathways using proteomics, MudPIT (multidimensional protein identification technology) as the main methods for analysis. In general, this area of research is very promising and interesting for a wide range of readers of this journal, as the authors present new data in the field of cancer research, as well as everything connected with it. However, before the article can be accepted, the authors should make minor adjustments to the text of the article that are necessary for a better perception of the material by readers.
1. The authors should provide more details about the methods they used to identify 2271 protein groups in the unfolding normal and cancer proteomes, as well as how exactly they separated and classified into groups and classes.
2. If possible, the authors should include the diagrams presented in Figures 6 – 9 in the appendix or try to combine the data presented on them into more comprehensive forms.
3. Analysis of the data in Figure 5 should be carried out taking into account the specified division into quadrants (Q1-Q4), as well as taking into account the weight shares of the presented data.
4. The data presented in Figure 1 are uninformative; there are no clear indications of exactly how subgroups and classes are divided; in this regard, the authors should either provide more details or provide an expanded legend for this figure.
5. A note on Figure 3. Why does one of the two figures show measurement errors, but not the second? What is this connected with?
